# A Mathematical Model for a Conceptual Design and Analyses of UAV Stabilization Systems

**Vadim Kramar** [1,*] **, Aleksey Kabanov** [1] **and Sergey Dudnikov** [2]

1 Department of Informatics and Control in Technical Systems, Sevastopol State University, 299011 Sevastopol, Russia; kabanov@sevsu.ru

2 Department of National Technology Initiative, Sevastopol State University, 299011 Sevastopol, Russia; sydudnikov@sevsu.ru

\* Correspondence: kramarv@mail.ru

**Abstract:** This article considers the principle of constructing mathematical models of functionally complex multidimensional multiloop continuous–discrete UAV stabilization systems. This is based on the proposal for constructing a mathematical model based on the class of the considered complexity of the stabilization system-multidimensionality, multi-rating, and elasticity. Multiloop (multidimensional) UAV stabilization systems are often characterized by the control of several interconnected state elements and the existence of several channels for the propagation of signals and mutual connections between individual objects. This is due to the need not only to take into account the numerous disturbing factors (for example, wind) acting on the control object as well as the need to use several points of application of control actions. Additionally, an important point is the possible separation of the mutual influence of the roll and yaw channels of the UAV on its synthesis and analysis. For this purpose, a mathematical model has been constructed using a description in the form of transfer functions, and therefore, in the form of structural diagrams. The principle of obtaining transfer functions is shown to demonstrate additional dynamic constraints introduced by elastic deformations into the stabilization loop through gyroscopic devices and accelerometers. This will make it possible to formulate a methodology for analyzing the influence of aeroelastic constraints on the stabilization loop, which will allow developing approaches to formulate requirements for the effective placement of gyroscopes and accelerometers on the UAV. The proposed approach allows creating a complete system of analysis and synthesis tools for complex multidimensional continuous–discrete UAV stabilization systems.

**Keywords:** multi-rate system; multiloop system; UAV stabilization system; continuous–discrete control system; elastic links





## 1. Introduction

The stabilization system of an unmanned aerial vehicle (UAV) is a set of devices and algorithms implemented in onboard computers that ensure the stable movement of the UAV's center of mass. The complexity of the synthesis of the UAV stabilization system requires an adequate mathematical model for its description.

Usually, the stabilization system of a UAV is a multiloop (multidimensional) system with many controllable parameters and input actions of more than one [1]. Multiloop UAV stabilization systems are characterized by the control of several interconnected state elements. The existence of several channels for the propagation of signals and mutual connections between individual objects is common [2,3].

This is due to the need not only to take into account the numerous disturbing factors acting on the control object, like the wind, but also the need to use several points of application of control actions [4].

A multiloop UAV stabilization system has a vector nature character of control actions, disturbing actions, control system parameters, and outputs [3,5].

Multiloop UAV stabilization systems are complex dynamic systems. Analytical methods for calculating their dynamic characteristics are very difficult. Therefore, structural solutions for these systems are of great importance. To obtain the structural solutions, special mathematical models that allow solving the assigned problems of analysis and synthesis must be built [1,6].

Currently, in the practice of analysis and synthesis of multiloop UAV stabilization systems, two approaches to the problem of obtaining mathematical models of such systems have been developed.

First approach: Taking into account the vector nature of the connections between the functional elements of the stabilization system, the vector–matrix representation of the equations describing the UAV stabilization system is used for constructing the mathematical models [7,8].

Second approach: A multiloop UAV stabilization system is considered a multiconnected set of dynamic links and is presented in the form of a structural diagram or an oriented graph [7,9].

In the first approach, there is a division of mathematical models into two groups [10–12]:

- Mathematical models in the time domain: These models are based on the vector–matrix form of representation of systems of differential equations and systems of finite-difference equations, the wide use of concepts, and methods of the state-space theory.
- Mathematical models based on the use of the Laplace transform and $z$-transform [13].

The obvious state of the modern stage of technical development is the transition from continuous automatic control systems to discrete systems. This is due to the use of microprocessors in control systems, including the circuits of UAV stabilization systems.

There are two main directions of using computers in stabilization systems: the development of a control program, where the results of calculations on computers are used as setting influences; and the use of computers directly in a closed loop to stabilize the control system and generate control signals.

The use of onboard computers to control UAV leads to the need to develop multiloop systems that contain the impulse elements with different quantization in sampling periods. The sampling period depends on the sampling rate of the signal sensors through the polling link, which depends on the way the data are read. In this case, the interrogation of sensors for measuring physical quantities (measuring points) is carried out separately when the state of the controlled process requires it. The selection of the polling cycle and response time depends on the dynamics of the process and the functions performed by the onboard computer. The shorter the cycle time is, the more accurate and complete the obtained information is. However, if this time is chosen to be too short, then the computer load increases significantly. With a long readout time, some information is lost, so the process may become unstable. When choosing a sampling period, a compromise must be found between both possibilities.

The complexity of control processes caused by the effects of quantization, and the presence of continuous and discrete elements leads to qualitatively new phenomena in the behavior of systems. Currently, two approaches are used to describe the continuous–discrete systems: continuous and discrete. In the first case, analysis and synthesis are carried out in the continuous domain and the obtained synthesis results are discretized. With the emerging approximation of continuous systems, the potential control capabilities are narrowed. In the second approach, the system is considered in a discrete domain, which leads to qualitative changes.

Each of these approaches leads to methodological errors since they are associated with the replacement of a continuous–discrete system with either a continuous or discrete model, each of which differs from the original one. The apparatus of the formal description of the process becomes much more complicated when trying to expand the class of systems under study. It is not possible to synthesize a high-precision stabilization system for such a complex control object as a UAV since it performs a flight task using conventional methods

for a long time due to the computational complexity of the algorithms that implement these methods and the lack of adequate models for describing it.

As noted, the use of onboard computers or separate digital devices in the measurement channels along with the devices of a continuous principle of operation leads to the combination of continuous and discrete processes in time [14].

In the given system, the formation of the digital vector of measurements is carried out using digital sensors. The digital vector of measurements is defined at a discrete set of the time $\Theta = \{ t_k | \ t_k = t_0 + kT; \ \ T > 0; \ \ k = 0, 1, 2, \ldots \}$ and consists of $n$ independent variables. Each component of the measurement vector is measured with its sampling period $T_i$, which is a multiple of the sampling period of the entire control system $T$, $T_i = m_i T$, $i = 1, 2, \ldots$.

For many measurement vector variables, the sampling periods can be the same. Measurements are made at times that belong to the specified discrete set $\Theta$, and at time intervals $t \in [t_k, \ t_k + T_i)$, these measurements are taken to be equal to a constant value.

For the mathematical description of such stabilization systems, it is impossible to apply purely continuous or purely discrete models. For these systems, the transition to a homogeneous description in the form of finite-difference equations is also impossible, especially for sufficiently large sampling periods in the discrete subsystems.

A complete description of UAV stabilization systems with a continuous–discrete nature of the processes is based on the use of mixed continuous–discrete models, which are a system of interrelated heterogeneous mathematical means of description: differential equations, finite-difference equations, differential equations with jumps, etc. [15].

One of the effective means of ensuring the stability of UAV stabilization systems is the rational placement and orientation of the sensitive elements of the control system, including gyroscopes and accelerometers. When using special meters, it is necessary to determine their location in the UAV body. The housings of modern aircraft-type UAVs, as a rule, are thin-walled elongated cylindrical structures, the tones of elasticity of which are commensurate with the eigenfrequencies of the UAV as a rigid body. The suppression of elastic vibrations, which are often the cause of system instability, is one of the most difficult problems in UAV design. The elasticity of the UAV body structure adversely affects the stability and accuracy of the stabilization system. Therefore, when synthesizing a UAV stabilization system, it is necessary to substantiate the effective arrangement of measuring devices on the UAV body.

Constructive methods for stabilizing the elastic vibrations of the UAV body lead to a complication of the UAV design or an expansion of the instrumental composition. All this, in turn, leads to an increase in the dry weight of the UAV. Therefore, in practice, algorithmic methods are widespread, which makes it possible to change the frequency characteristics of the stabilization loop. The considered problems are solved by approximate methods, with significant assumptions, since building accurate models is currently a laborious process. Thus, an important and urgent task is the development of adequate problem-oriented mathematical models, which are the basis for the synthesis and analysis of the UAV stabilization system, taking into account the maximum number of influencing factors.

In this article, we will consider various approaches to the construction of mathematical models of the stabilization system, considering the influence of various factors to form the methods for the synthesis of these systems in the full combination of all elements of complexity according to the principle "from simple to complex".

## 2. Methods

### 2.1. Classes of Mathematical Models of Continuous–Discrete Systems

If discrete subsystems operate on the same time sequence, then such a continuous–discrete automatic control system is called single-rate [16]. In [17–20], such a linear automatic control system is described by a set of differential equations describing the control

system on intervals of continuity $t_k \leq t < t_{k+1}$, and difference equations describing state jumps at discrete times $t_k$:

$$\left. \begin{array}{l} \frac{dx(t)}{dt} = A_1(t)x(t) + B_1(t)u_1(t), \quad t_k \leq t < t_{k+1}, \\ x(t_k) = A_2(t_k)x(t_k - 0) + B_2(t_k)u_2(t_k), \quad t_k \in \Theta, \quad x(t_0) = x_0 \end{array} \right\}, \tag{1}$$

where $A_1(t)$, $B_1(t)$ are the matrices of the coefficients of the continuous part of the system, $u_1(t)$ is the continuous control vector, $A_2(t_k)$, $B_2(t_k)$ are the matrices of the coefficients of the discrete part of the system, $u_2(t_k)$ is the vector of the discrete control, $x(t)$ is the $n$-dimensional vector of the state of the stabilization system, the elements of which are continuous functions on finite time intervals $[t_k, \ t_{k+1}]$ with discontinuities of the first kind at times $t_k \in \Theta$, $x(t_k - 0)$—the value of the state vector at the moment of action of the pulse quantizer:

$$\Theta = \left\{ t_k, \ k = 1, 2, \ \ldots, \ \left| \inf_k (t_{k+1} - t_k) > 0 \right| \right\}, \tag{2}$$

—the set of times $t_k$ at which the state vector $x(t)$ undergoes discontinuities.

The solution to systems of Equation (1) is piecewise continuous functions that satisfy the differential equations of the system at finite time intervals $[t_k, \ t_{k+1}]$, and at times from the set $\Theta$ undergo discontinuities of the first kind following the given finite difference equations.

In many problems, discrete subsystems of continuous–discrete UAV stabilization systems operate on different time sequences. This is due to both the functional features of discrete subsystems and the need to achieve their maximum efficiency. Therefore, for example, in an onboard computer of an aircraft-type UAV, signals from angular velocity sensors are processed much more often than signals about the position of the center of mass.

It is shown in [16] that the general model described by the system (1) allows one to describe the dynamics of multitasking automatic control systems since the time sequence in it is arbitrary. To explicitly highlight the subsystems and take into account the peculiarities of the dynamics of the UAV stabilization system, it is advisable to introduce a mathematical model of multi-rate continuous–discrete control systems.

The general model of a multi-rate continuous–discrete stabilization system in the time domain can be represented as

$$\left. \begin{array}{l} \frac{dx_C(t)}{dt} = A_{CC}(t)x_C(t) + \sum\limits_{i=1}^{N} A_{CD_i}(t)x_{D_i}([t])_i + B_{CC}(t)u_C(t), \quad t \notin \Theta, \\ x_{D_i}([t])_i = A_{CD_i}([t]_i)x_C([t]_i) + \sum\limits_{j=1}^{N} A_{DD_{ij}}([t]_i)x_{D_j}([t - 0]_j) + \\ \qquad + B_{DD_i}([t]_i)u_{DD_i}([t]_i), \qquad t \in \Theta, \quad i = 1 \ldots N; \\ x_C(t_0) = x_{C0}; \qquad x_{D_i}(t_0) = x_{D_{i0}}, \qquad i = 1 \ldots N. \end{array} \right\}, \tag{3}$$

In system (3), the following designations are accepted: $x_H(t)$—the state vector of the continuous part of the control system; $x_{D_i}(t)$—the state vectors of the discrete subsystems of the stabilization system; $A_{CC}(t)$, $B_{CC}(t)$—the matrices of the coefficients of the continuous part of the control system; $A_{DD_i}(t)$, $B_{DD_i}(t)$—the matrices of the coefficients of the discrete subsystems of the automatic control system; $A_{CD_i}(t)$—the matrix of the coefficients of the discrete part of the system affecting the continuous; $u_{CC}(t), \ldots, \ u_{DD}([t]_i)$—the current time functions:

$$\left. \begin{array}{l} ki(t) = \max\left\{ k \mid t_k^i \in \Theta_i, \ t_k^i \leq t \right\}, \\ [t]_i = t_{ki(t)}, \quad \{t\}_i = t - [t]_i. \end{array} \right\}, \tag{4}$$

$$\Theta_i = \left\{ t_1^i, \ t_2^i, \ t_3^i, \ \ldots \mid t_{k+1}^i - t_k^i \geq T_i > 0 \right\}, \quad i = 1, \ldots, N, \tag{5}$$

—a set of moments in time that determine the functioning of discrete subsystems of UAV stabilization systems, where:

$$t_k^i = kT_i, \quad T_i = const, \tag{6}$$

and $T_i$ is the sampling period of the *i*-th discrete subsystem of the UAV stabilization system.

The simplest case is when the sampling periods of discrete subsystems (6) are mutually rational numbers, which is most often encountered in practice. For example, if the discrete subsystems of the stabilization system are implemented in the form of programs of one control computer, then their discreteness cycles contain an integer number of sampling periods of the computer's clock generator, and therefore, are mutually rationally simple.

A great development in the issue of constructing mathematical models of multiloop but not multidimensional control systems with different sampling periods specified using transfer functions (based on the z-transform and discrete Laplace transform), despite the great complexity of the proposed approaches and serious restrictions on their application, was achieved thanks to works [13,21,22]. This approach is very relevant due to the ability to work with models in the form of structural diagrams.

In [23,24], a mathematical model of a multiloop multidimensional system is obtained in the operator vector–matrix form, which takes into account the influence of all quantizers of the system. This approach is fully applicable to the description of the UAV stabilization system, which, as mentioned earlier, is a multiloop multidimensional continuous–discrete system. For completeness, we present the construction of these mathematical models [23,24].

Let $u(s)$ be the vector of control actions on the object of $m \times 1$ dimension, and $y(s)$—the vector of outputs of the object of $p \times 1$ dimension. Let $x(s) = [x_1(s), x_2(s), \ldots, x_r(s)]^T$ be a vector of variables quantized on an analog-to-digital converter with sampling periods $T_1, T_2, \ldots, T_r$, respectively (among which there may be equal ones). We will consider commensurate sampling periods $T_1, T_2, \ldots, T_r$ that are multiples of a certain sampling period $T$, i.e., $T_1 = n_1 T$, $T_2 = n_2 T$, ..., $T_r = n_r T$, where $n_1, n_2, \ldots, n_r$ are natural numbers.

The equations of a continuous object and analog circuits of the stabilization system from the object to the quantization keys have the form:

$$y(s) = W_0(s)u(s),$$
$$x(s) = E(s)y(s) + B(s)u(s), \tag{7}$$

where $W_0(s)$, $E(s)$, $B(s)$ are the matrices of transfer functions of the corresponding dimensions. Thus:

$$x(s) = U(s)u(s) = [E(s)W_0(s) + B(s)]u(s). \tag{8}$$

Let $x^*_{1T_1}(s)$, $x^*_{2T_2}(s)$, ..., $x^*_{rT_r}(s)$—discrete Laplace transforms quantized concerning the sampling periods $T_1, T_2, \ldots, T_r$ of the variables $x_1(t)$, $x_2(t)$, ..., $x_r(t)$, respectively. Each of the quantized signals $x_i(k_i T_i)$, $k_i = 0, 1, \ldots$, $i \in \overline{1, r}$ is converted by a corresponding digital circuit and summed with other similar signals, as a result of which a control action is formed. In addition to digital circuits, analog circuits (from the outputs of the object) can also be used in the formation of control actions. Let us write the equation for the "*k*-th" component of the vector of control actions:

$$u_k(s) = -\sum_{i=1}^{r} d_{ki}(s)x^*_{iT_i}(s) - \sum_i^p f_{ki}(s)y_i(s) + u_{kz}(s). \tag{9}$$

where $d_{ki}(s)$, $f_{ki}(s)$ are the transfer functions of parallel digital and analog circuits, and $u_{kz}(s)$ is the reference action generated by the digital circuit.

With the digital summation of converted digital signals (with sampling period $T$ and corresponding signal repetition) and subsequent digital-to-analog conversion, it can be assumed that:

$$d_{ki}(s) = \frac{1 - e^{-Ts}}{s} W^*_{kiT_i}(s). \tag{10}$$

where $W^*_{kiT_i}(s)$ is the periodic (with a period $\frac{2\pi j}{T_i}$) transfer function of the digital (impulse) conversion, and:

$$u_{kz}(s) = \frac{1 - e^{-Ts}}{s} u^*_{kzT}(s), \tag{11}$$

where $u^*_{kzT}(s)$ is the Laplace transform of the quantized reference action–periodic (with a period $\frac{2\pi j}{T}$) function.

In a more general case of combined digital and analog (using other signals) summation, it is also possible to accept relation (10) to describe digital feedback circuits with a corresponding complication of the periodic part of the transfer function, for example, by writing:

$$d_{ki}(s) = \frac{1 - e^{-T_i s}}{s} W^*_{kiT_i}(s) = \frac{1 - e^{-Ts}}{s} \cdot \frac{1 - e^{-T_i s}}{1 - e^{-Ts}} W^*_{kiT_i}(s) = a(s)\widetilde{W}^*_{kiT_i}(s),$$

where $a(s) = \frac{1 - e^{-Ts}}{s}$, $\widetilde{W}^*_{kiT_i}(s)$—periodic part of the transfer function $d_{ki}(s)$.

Let us introduce a matrix $W_*(s)$ with elements $W^*_{kiT_i}(s)$ and vectors $x_*(s)$, $v^*(s)$ with elements $x^*_{iT_i}(s)$ and $u^*_{kzT}(s)$. The symbol "*" at the bottom marks the property of periodicity of the matrix $W_*(s)$ and the vector $x_*(s)$ for the corresponding periods. At the same time, all these elements satisfy relations of the form $x(s + 2\pi j/T) = x(s)$.

Taking into account relations (10) and (11) as rather general and writing down the equations for control actions in vector–matrix form:

$$u(s) = -a(s)W_*(s)x_*(s) - F(s)y(s) + a(s)v^*(s),$$

in which $F(s)$ is a $m \times p$ matrix with elements $f_{ki}$, we obtain, taking into account relations (7) and (8):

$$u(s) = -G(s)W_*(s)x_*(s) + G(s)v^*(s), \tag{12}$$

$$x(s) = -C(s)W_*(s)x_*(s) + C(s)v^*(s), \tag{13}$$

$$y(s) = -L(s)W_*(s)x_*(s) + L(s)v^*(s), \tag{14}$$

where $G(s) = a(s)(I + F(s)W_0(s))^{-1}$, $C(s) = U(s)G(s)$, $L(s) = W_0(s)G(s)$.

Relations (12), (13), (14) together with the vector $x_*(s) = \begin{bmatrix} x^*_{1T_1}(s) & x^*_{2T_2}(s) & \dots & x^*_{rT_r}(s) \end{bmatrix}^T$ represent a model of the closed multiloop, multidimensional multi-rate continuous–discrete UAV stabilization system, which takes into account both the influence of all quantizers of the system and its multiloop, including the cross-connections of control channels. Note, as indicated above, that the use of mathematical models of automatic control systems in the form of transfer functions allows using models in the form of structural diagrams.

## 2.2. Full Continuous–Discrete Multi-Rate Model of the UAV Stabilization System

The structural diagram of the elastic UAV lateral motion stabilization system is shown in Figure 1. An analog of the considered system is given in [25]. In this figure, $\psi$—yaw angle; $\gamma$—roll angle; $\delta_N$ and $\delta_E$—angles of deflection of rudders and ailerons, respectively; $\varepsilon_N$ and $\varepsilon_E$—stabilization errors. Stabilization dynamics are determined by transfer functions $\Phi_{ST}(s)$—steering track; $W_{KL}(s)$—kinematic link; $W_{GT}(s)$—gyrotachometer; $W_e(s)$—extrapolator; and transfer functions of the control object (excluding and taking into account elastic connection).

All the indicated transfer functions are rational except for $W^y_{\delta_N \psi}(s)$, $W^y_{\delta_N z}(s)$—meromorphic functions with purely imaginary simple poles, which describe elastic constraints.

The transfer functions of the control object (a rigid vehicle in rotational motion around the center of mass) can be determined by transforming, according to Laplace, the system of first-order differential equations describing the rotational motion of the aircraft around the center of mass in the horizontal plane:

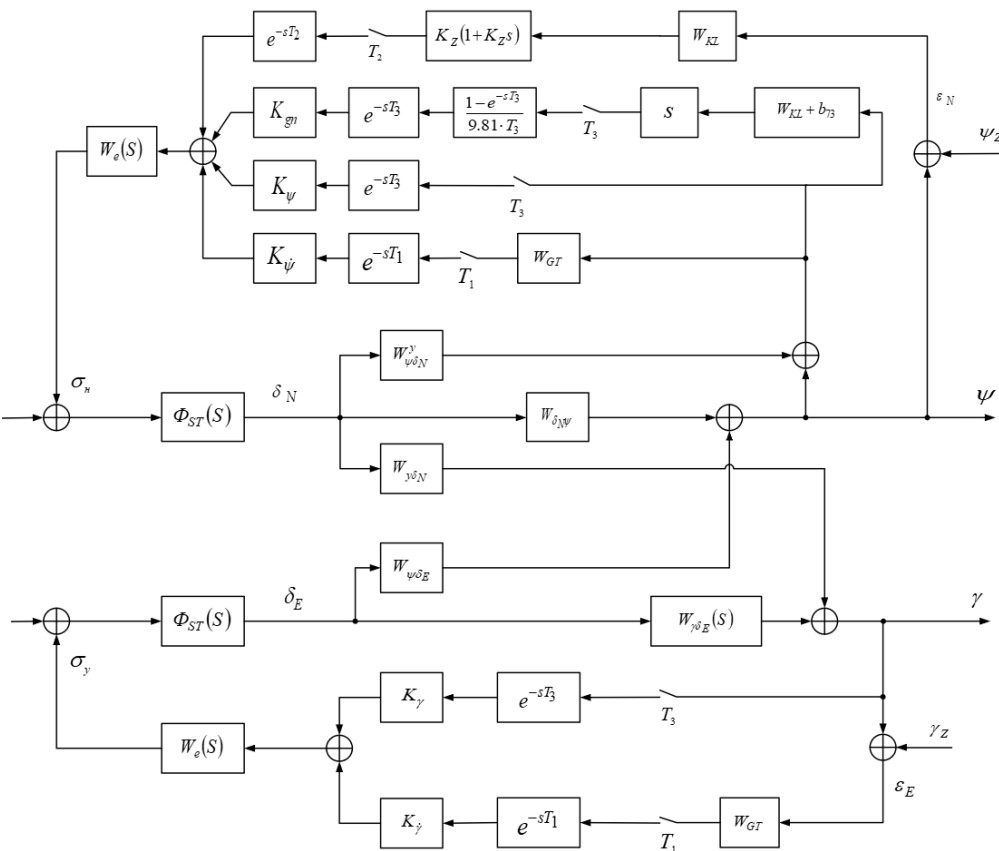

**Figure 1.** Block diagram of the complete model of the lateral system stabilization UAV.

$$\begin{cases} \dot{\beta} = b_{11}\beta + b_{12}\omega_x + b_{13}\omega_y + b_{16}\delta_N + b_{14}\gamma \\ \dot{\omega}_x = b_{21}\beta + b_{22}\omega_x + b_{23}\omega_y + b_{26}\delta_N + b_{27}\delta_E \\ \dot{\omega}_y = b_{31}\beta + b_{32}\omega_x + b_{33}\omega_y + b_{36}\delta_N + b_{37}\delta_E \\ \dot{\gamma} = \omega_x + b_{43}\omega_y \\ \dot{\psi} = \omega_y \end{cases} \qquad (15)$$

In these equations: $\beta$—the slip angle; $\omega_x$, $\omega_y$—the projection of the angular velocity vector on the axis $x$ and $y$; $b_{ij}$—aerodynamic coefficients [25].

### 2.3. Influence of Deformations of the UAV Body on Its Dynamics

The transfer functions of elastic links $W^y_{\delta_N\psi}(s)$, $W^y_{\delta_N z}(s)$ characterize the effect of the dynamics of stabilization of elastic vibrations of a moving UAV. This effect can be quite noticeable.

All mechanical rod structures, including the UAV body, are not rigid structures that vibrate [26,27]. The main reason that causes bending and flexural vibrations of the body is the control torque generated by the steering elements. This becomes especially important for pumped aircraft because such objects tend to be lighter in weight. To increase the UAV's flight range, the most acceptable solution is to increase its length, which in turn increases the flexibility of the body. The elasticity of the UAV body structure harms the stability and quality of the control system.

A large number of studies have been devoted to the study of flying vehicles as elastic bodies, for example [28].

In [29], it is shown how oscillations affect the motion of an aircraft. For example, under the influence of a disturbing moment $M_z$, the aircraft is deflected by a certain yaw angle $+\psi$. Then, the stabilization machine, turning the steering elements $(+\delta_1)$, will create a control torque $M_\delta$, which should compensate the harmful effect of the disturbance (see Figure 2).

However, the body of the aircraft, as already mentioned above, is not an absolutely rigid structure, and it will bend under the action of the torque $M_\delta$, as a result, the yaw angle sensor, placed on the gyro platform, will register the angle not the angle $\psi$, but $\psi + \Delta\psi_{fl}$ (see Figure 3).

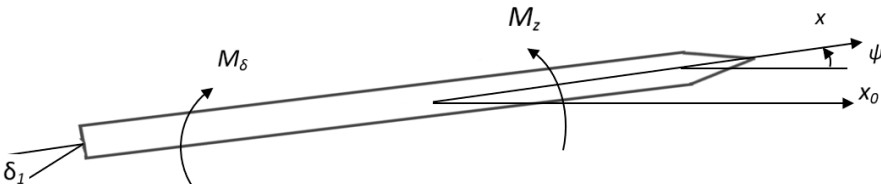

**Figure 2.** Action control and disturbing moments.

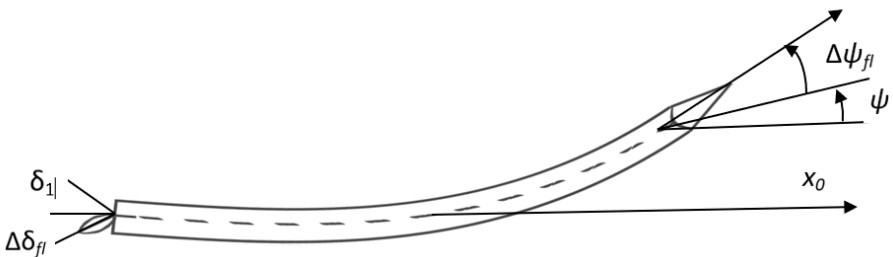

**Figure 3.** Deflection of the aircraft under the control torque.

For the analysis of oscillations, the transfer function of the UAV is obtained as an object of regulation, taking into account the oscillators, which is not always convenient, since they are usually satisfied with approximate models of the phenomenon in the form of several vibrational links corresponding to the fundamental tones of elastic vibrations. The question of the accuracy of such approximations of the model is solved mainly by experimental means in each specific case. In this regard, there is a problem with constructing an accurate dynamic model of an elastic link under certain assumptions. This will make it possible to compare the exact and approximate characteristics of the elastic link and, therefore, reasonably introduce the corresponding simplifications.

The elastic properties of a UAV can be significantly manifested in its movement dynamics. Transient processes in the stabilization loop, occurring under the action of aerodynamic forces, are accompanied by the elastic deformations of the UAV body, which affect the signals of the measuring devices. For example, the bending angles of the body add an additional component to the signals of gyroscopes, and the accelerations of elastic vibrations are manifested in the signals of accelerometers. Due to this, additional feedbacks–elastic links appear in the stabilization loop, which should be taken into account when analyzing the properties of the loop. The mathematical description of these links, as well as the subsequent analysis, taking into account the stabilization loop, is a very difficult task. Usually, they are satisfied with approximate models of the phenomenon in the form of one or more vibrational links corresponding to the fundamental tones of elastic vibrations. Below, there is a description of the principle of constructing a mathematical model that describes an elastic link in the form of transfer functions.

Methods for constructing transfer functions are given in [30–32]. Furthermore, when describing the construction of a mathematical model of the UAV stabilization system, taking into account elastic vibrations, we will briefly give a method for constructing transfer functions describing an elastic link.

A model of the stabilization system, the block diagram of which is shown in Figure 1 will be called "the complete model". Sampling periods are assumed to be commensurate, i.e., multiples of a certain number $T$—the greatest common divisor of sampling periods. Otherwise, the problem of analyzing the model becomes rather complicated, and as been mentioned earlier, practically insoluble. However, usually, due to the operation of one

control computer onboard the UAV, the condition of the frequency of sampling periods is satisfied. "The complete model" is the basic model for obtaining a family of simplified models of the UAV stabilization system.

*2.4. Hierarchical Models of the UAV Stabilization System*

It is natural to develop methods for the synthesis of multiloop multi-rate continuous–discrete UAV stabilization systems according to the principle "from simple to complex." The degree of complexity is determined by the degree of complexity of the model. Therefore, it is natural to introduce into consideration a certain hierarchy of models with varying degrees of simplification obtained from the complete model under various kinds of assumptions. Assumptions apply to those aspects of the overall model that determine the complexity of the synthesis and analysis problem.

The meromorphism of the transfer functions $W^y_{\delta_H \psi}(s)$, $W^y_{\delta_H z}(s)$ greatly complicates the task of synthesizing and analyzing the system. If elastic links are described approximately in the class of rational functions, a simplified model follows from the complete model, which we will call "model 1".

Disregarding the effect of quantization in digital feedback loops of the full model results in "model 2". By separating the angular motion stabilization system from it, we obtain a block diagram, which is shown in Figure 4. "Model 2" is a continuous two-dimensional stabilization system taking into account the elastic links of the object. If in "model 2" we neglect the mutual influence of the heading and roll channels, we obtain a continuous one-dimensional model of the stabilization system. We will call it "model 3". The block diagram of "model 3" is shown in Figure 5. Finally, neglecting the elastic constraints in "model 2", we obtained "model 4". Its structural diagram is identical to the structural diagram shown in Figure 4, at $W^y_{\delta_H \psi}(s) = 0$. Synthesis and analysis of complex multiloop UAV stabilization systems with specific signs of complexity are carried out following the given hierarchy of models.

Let us consider the application of the proposed approach, for example, for analyzing the stability of the UAV stabilization loop. By "model 3", the stabilization circuit of a "rigid" unmanned aerial vehicle can be considered in the form shown in Figure 6. In this figure, $\psi$—the yaw angle of the vehicle; $\delta$—the rudder deflection angle; $\psi_z$—the specified yaw course angle; and $\varepsilon$—the stabilization error.

The dynamics of stabilization are determined by the transfer functions of $W_0(s)$—the control object (UAV in rotational motion around the center of mass); and $W_{oc}(s)$—the feedback loop of the stabilization system. When small elastic deformations occur that do not change the aerodynamic forces of the UAV occur, the transfer function $W_{oc}(s)$ of the object remains the same, but the stabilization contour changes as new connections appear. If the sensitive element of the feedback circuit is a gyroscopic device that responds to the yaw angle, which characterizes the position of the vehicle body relative to the constant direction, then the modified contour will have the form shown in Figure 7.

In this figure, $W_y(s)$ is the transfer function of the elastic connection of the object, determined by the equality $\mu(s) = W_y(s)\delta(s)$, $\delta(t)$—rudder angle.

To determine the transfer function, we note that, according to the equations of motion of a rigid apparatus, it is possible to establish the relationship in the images between the angle of sliding of the apparatus and the angle of deflection of the steering wheel:

$$\beta(s) = C(s)\delta(s), \tag{16}$$

where $C(s)$ is the corresponding transfer function of the "rigid" UAV.

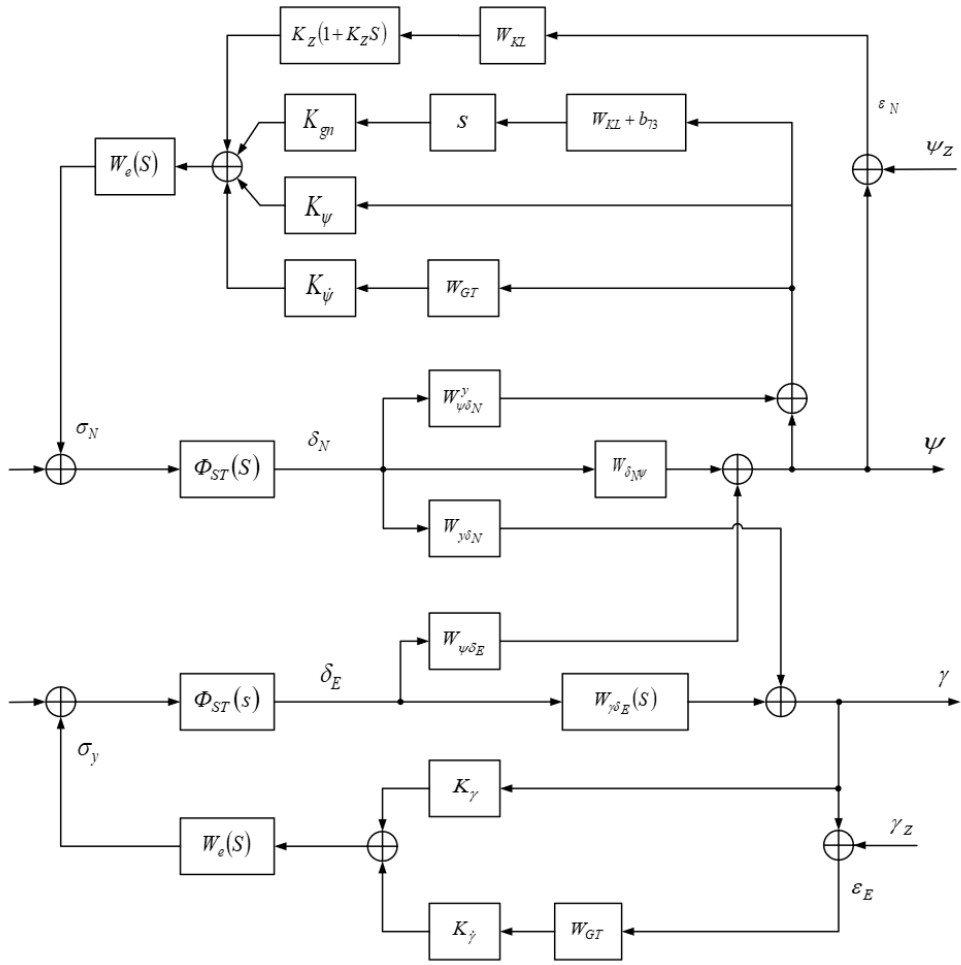

**Figure 4.** Block diagram of a continuous two-dimensional lateral stabilization system taking into account the elastic connections of the object ("model 2").

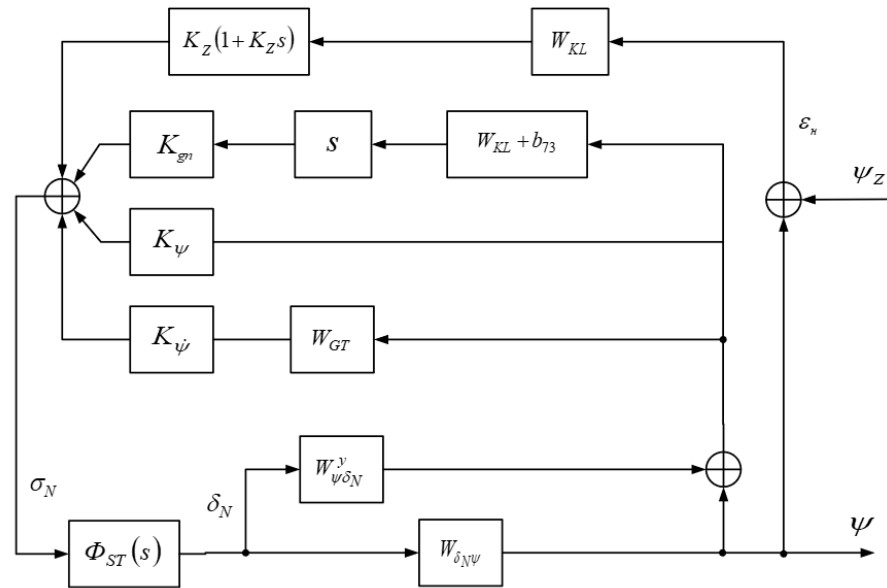

**Figure 5.** Block diagram of a one-dimensional model of the course stabilization system ("model 3").

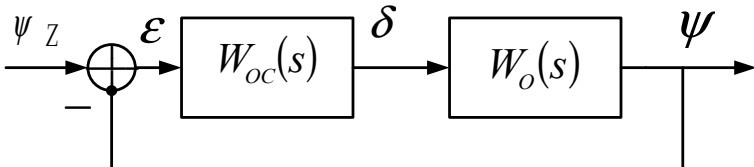

**Figure 6.** "Rigid" vehicle course stabilization contour.

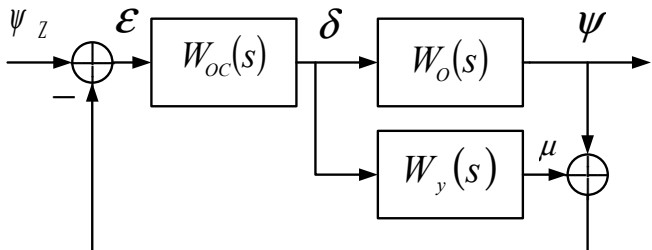

**Figure 7.** Modified UAV course stabilization contour.

In this case, we take into account that in the presence of a free gyroscope, an additional signal of this sensitive element arises, since it will register the bending angle of the apparatus body. For small vibrations, this angle can be defined as

$$\mu(x,s) = \left.\frac{\partial}{\partial x}Z(x,s)\right|_{x=x_G}, \tag{17}$$

where $x_G$ is the abscissa of the gyroscope attachment point.

Differentiating by $x$ the expression for the deflection, we find:

$$\mu(x,s) = V^{\beta}(x,s)\beta(s) + V^{\delta}(x,s)\delta(s), \tag{18}$$

where $V^{\beta}(x,s) = \frac{\partial}{\partial x}U^{\beta}(x,s)$, $V^{\delta}(x,s) = \frac{\partial}{\partial x}U^{\delta}(x,s)$, $\beta(t)$—sliding angle and:

$$Z(x,s) = U^{\beta}(x,s)\beta(s) + U^{\delta}(x,s)\delta(s),$$

represents the transfer function of the UAV as an elastic link.

Then, the ratio for the bending angle of the UAV body can be given the form:

$$\mu(x,\ s) = \left[V^{\beta}(x,\ s)C(s) + V^{\delta}(x,\ s)\right]\delta(s). \tag{19}$$

Fixing in this expression the value at the given position of the gyroscopic device in the UAV ($x = x_G$), and taking:

$$W_y(s) = V^{\beta}(x_G,\ s)C(s) + V^{\delta}(x_G,\ s) \tag{20}$$

we obtain the relation for $\mu(s)$, where $\mu(s) = \mu(x_G,\ s)$.

A more complex situation arises when the UAV stabilization system contains two gyroscopic devices located in different places—a free gyroscope and a gyrotachometer. In this case, each device reacts to the UAV bending angle in its place. The block diagram takes the form shown in Figure 8.

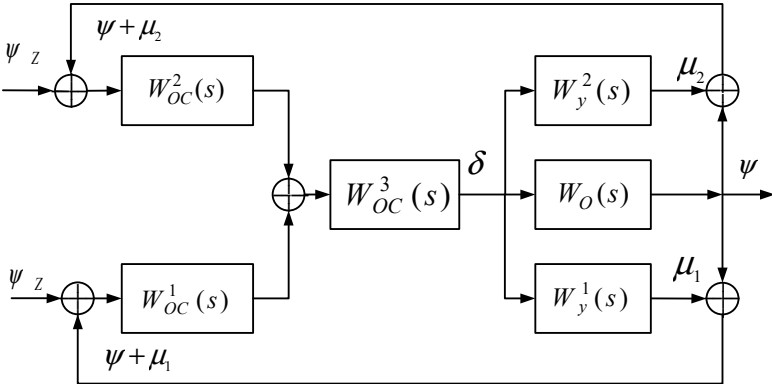

**Figure 8.** Block diagram of the stabilization system with two gyroscopic devices.

Here, each of the transfer functions $W_y^1(s)$, $W_y^2(s)$ is determined by an expression of the form (20) with the values of the abscissa $x_G^1$, $x_G^2$, respectively. If $x_G^1$, $x_G^2$ coincide, the elastic links are combined, and:

$$W_{oc}(s) = [W_{oc}^1(s) + sW_{oc}^2(s)]W_{oc}^3(s).$$

Similarly, we can consider the structural diagrams of the stabilization system when used in the system of accelerometers.

## 3. Results

### 3.1. Analysis of the Multi-Rate Multiloop UAV Lateral Stabilization System

We used the constructed models to analyze the UAV stabilization system. Let us compare the transient processes (response to wind) of the UAV stabilization systems for the different types of models mentioned above. We will consider the decoupled model without taking into account the influence of the roll. Figure 9 shows a Simulink—a diagram of a UAV course stabilization system described by a continuous system and a UAV course stabilization system described by a multiloop continuous–discrete system with the same parameters of transfer functions (Figure 3) and sampling periods $T_1$, $T_2$, $T_3$ of digital feedback circuits of control signals, which are multiples of a certain number $T$—the greatest common divisor of sampling periods.

The dynamics of stabilization is determined by the transfer functions: $\Phi_{ST}(s)$—of the steering tract (in Figure 9—$F_{rt1}$):

$$\Phi_{ST}(s) = \frac{W_{PT}(s)}{1 + W_{PT}(s)}, \tag{21}$$

$$W_{PT}(s) = k_{PT}W_{PM}(s)W_{\mu y}(s), \quad W_{PM}(s) = \frac{e^{-\tau_{PM}s}}{s(T_{PM}s + 1)}, \quad W_{\mu y}(s) = \frac{1}{T_{\mu y}s + 1} \tag{22}$$

$W_{KL}(s)$—of the kinematic link (in Figure 9—$W_{kz1}$):

$$W_{KL}(s) = a_{41}\frac{-\frac{b_{16}s^2}{\mu_2} + \frac{b_{16}b_{33}s}{\mu_2} + 1}{s\left(\frac{b_{36}s}{\mu_2} + 1\right)}, \tag{23}$$

$W_{GT}(s)$—of the gyrotachometer (in Figure 9—$W_{gt1}$):

$$W_{GT}(s) = \frac{s}{(T_{GT}^2 s^2 + 2\zeta_{GT}T_{GT}s + 1)(\tau_{GT}s + 1)}, \tag{24}$$

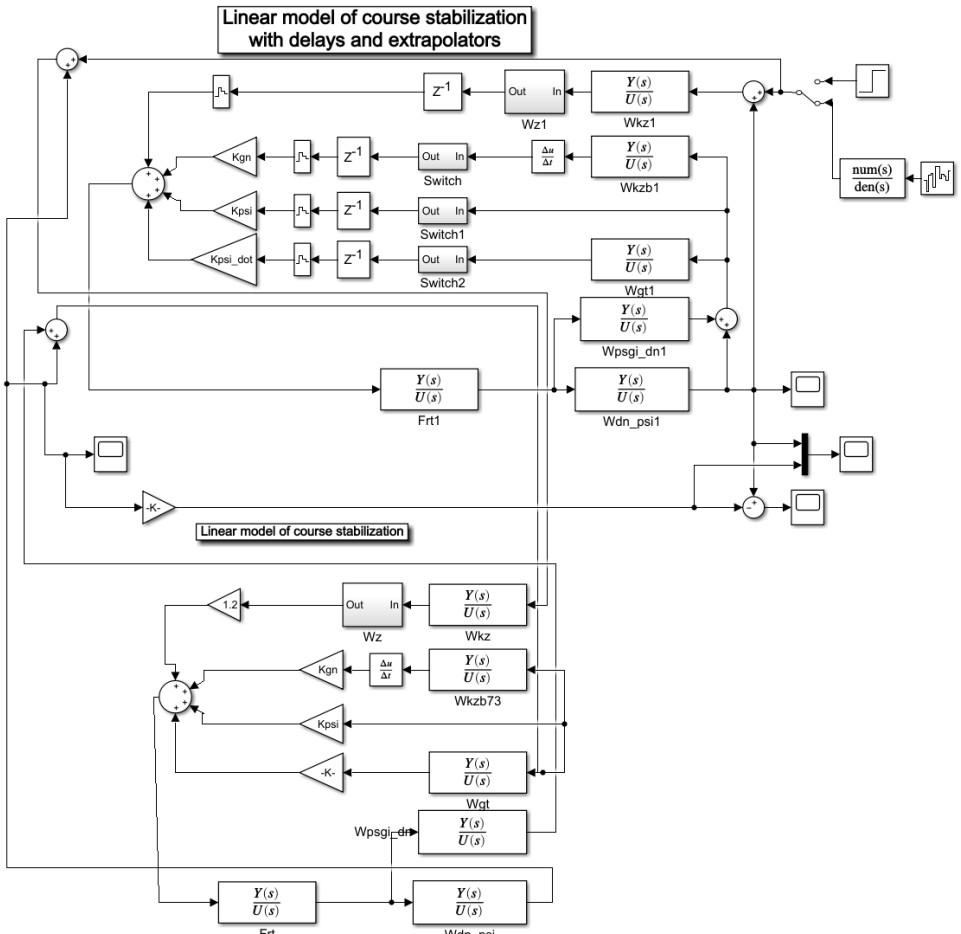

**Figure 9.** Simulink—diagram of a continuous and multiloop continuous–discrete UAV course stabilization system.

In the above ratios, $b_{ij}$—aerodynamic coefficients [25]; $T$ and $\tau$ with the corresponding indices—time constants.

The transfer functions of the control object (rigid apparatus in rotational motion around the center of mass):

$$W_{\delta_N \psi}(s) = \frac{\psi(s)}{\delta_N(s)}, \tag{25}$$

can be determined by transforming, using the Laplace transform, a system of first-order differential equations describing the rotational motion of the aircraft around the center of mass in the horizontal plane (15) (in Figure 9—Wdn_psi1).

Additionally, the model takes into account the transfer function $W_{\psi \delta_H}^{y}(s)$, which characterizes the effect on the dynamics of stabilization of elastic vibrations of a moving UAV (in Figure 9—Wpsgi_dn1).

The results of the deviation of the stabilization system from the zero value of the yaw angle are shown in Figure 10. The red (dashed) line indicates the transient process of the UAV heading stabilization system described by the continuous system, and the blue (solid) line—the transient process of the UAV heading stabilization system described by the multi-rate continuous–discrete system. We see that the first system has a higher overshoot.

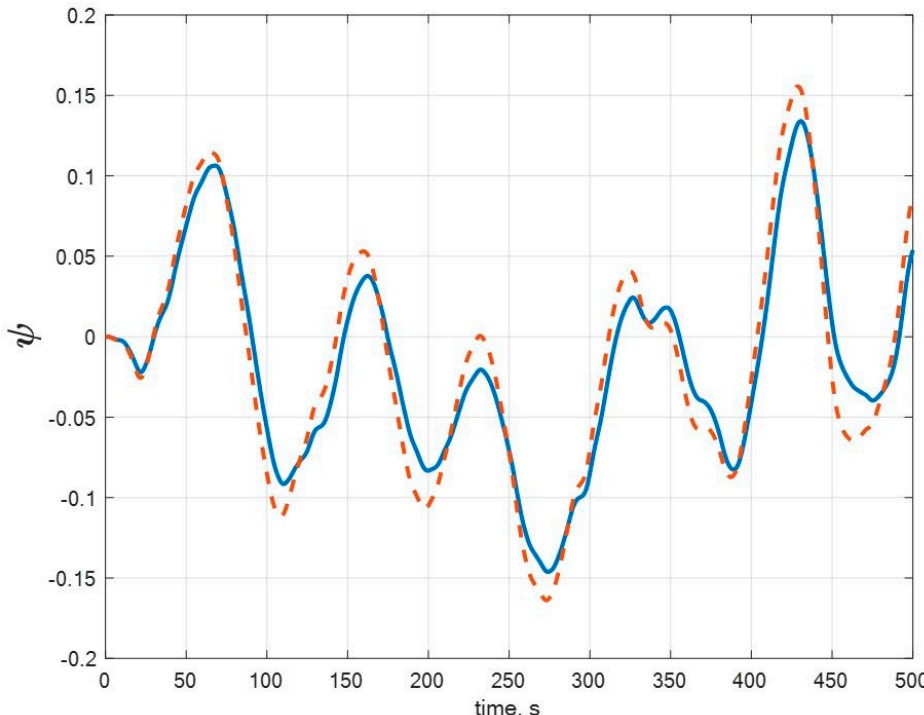

**Figure 10.** Transient processes of continuous and multi-rate continuous-discrete system of UAV course stabilization.

A similar situation was observed in the analysis of the UAV course stabilization system described by a single-rate continuous–discrete system and a UAV course stabilization system described by a multi-rate continuous–discrete system with the same parameters of transfer functions (see Figure 11). In Figure 11 red (dashed) line shows the transient process of a single-rate discrete UAV stabilization system with the sampling period $T_1$, and the blue (solid) line shows the transient process of the multi-rate continuous–discrete UAV stabilization system.

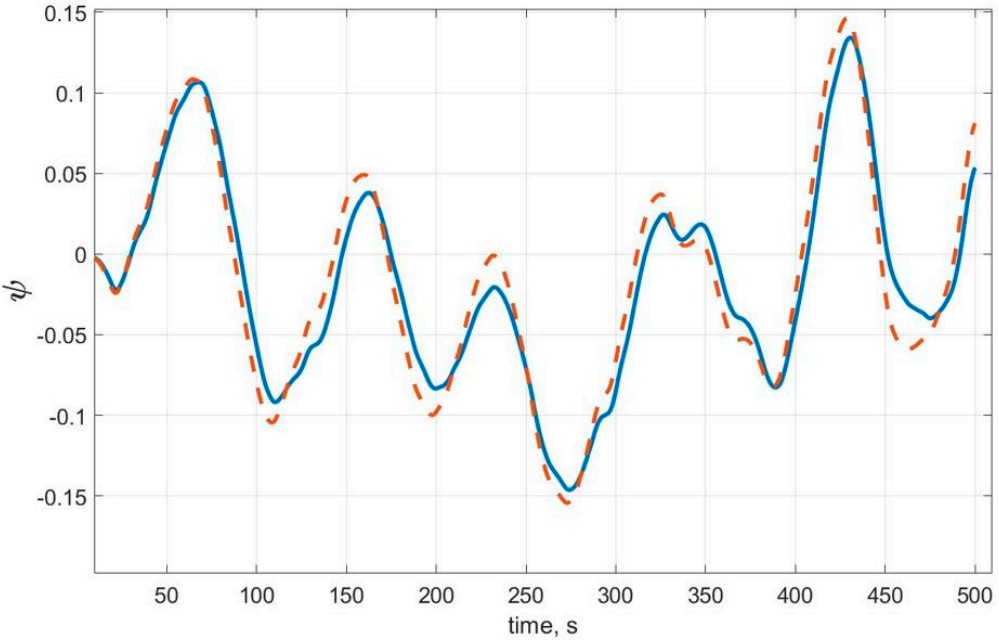

**Figure 11.** Transient processes of a single-cycle discrete system with a sampling periods $T_1$ and a multi-rate discrete UAV stabilization system.

Analyzing the behavior of systems, the Simulink models of which are shown in Figure 9, it can be concluded that the transient processes of systems built according to the approximate models (in our case, continuous and discrete), although they are similar, do not coincide with the transient processes of the full model. The more parameters are taken into account in the model, the more accurate the transition process becomes. This can be analyzed by analyzing the behavior of the quadratic integral error. We can conclude that the quadratic integral error for a model of a stabilization system in the form of a continuous system grows faster than for a model in the form of a multi-rate continuous–discrete system and the quadratic integral error for a model of a stabilization system in the form of a single-rate system grows faster than for a model in the form of a multi-rate continuous–discrete system, but at the same time, slower than for a UAV heading stabilization system described by a continuous system. It should also be noted that with an increase in the sampling period in the considered single-rate system, the stabilization system becomes unstable.

### 3.2. Analysis of the Elastic UAV Lateral Stabilization System

Now, let us consider the UAV stabilization system in the presence of the influence of elastic deformation. The Simulink model of the specified system is shown in Figure 12.

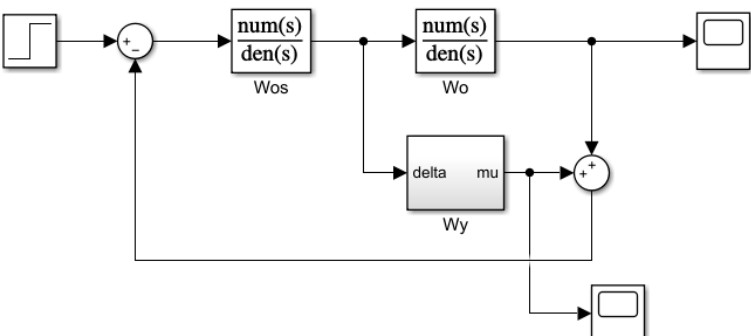

**Figure 12.** Simulink model of the system with the influence of elastic deformation on the object in Figure 9.

In this Simulink scheme (see Figure 12), the "Wy" block simulates the elastic deformation effect using the relationships given in Section 3.

Let us present the transient processes for the transfer function of the elastic link for UAVs with different masses and lengths, obtained using the scheme (see Figure 13). Figures 14 and 15 show the transient processes for a UAV with a length of 10 m, a mass of 500 kg and 1000 kg, respectively. It can be seen from the figures that the mass of the UAV significantly affects the type of the transition process. In this case, even a small deviation in the steady-state occurs. A similar situation of the influence of oscillations occurs when the length of the UAV changes.

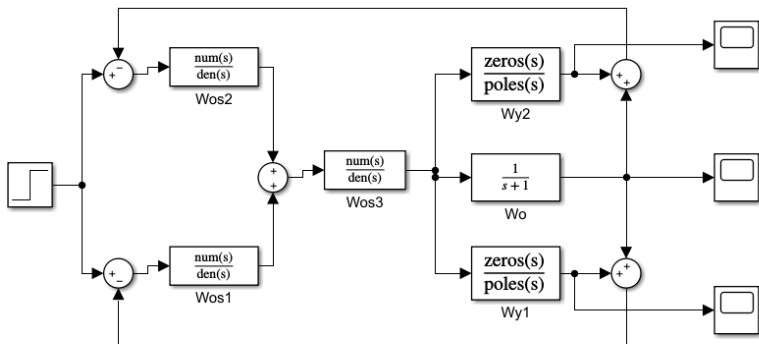

**Figure 13.** Simulink model of the system with the influence of elastic deformation on the object in Figure 10.

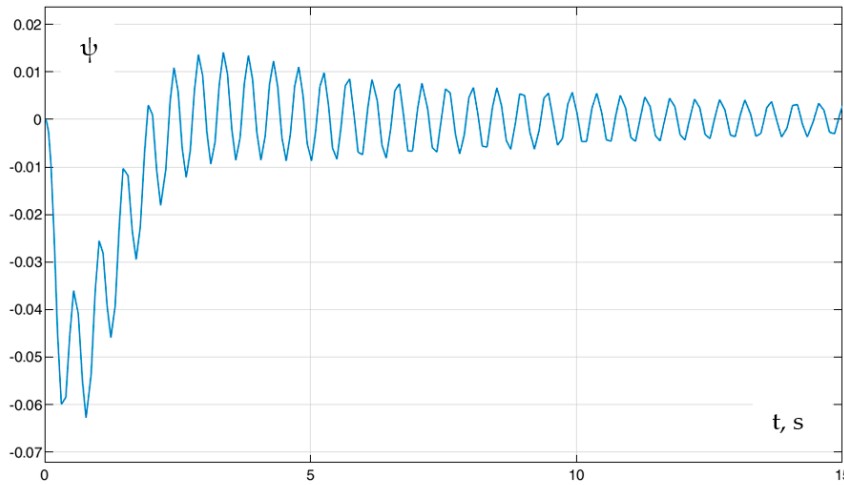

**Figure 14.** The transient process of an elastic link with a UAV mass of 500 kg and a length of 10 m.

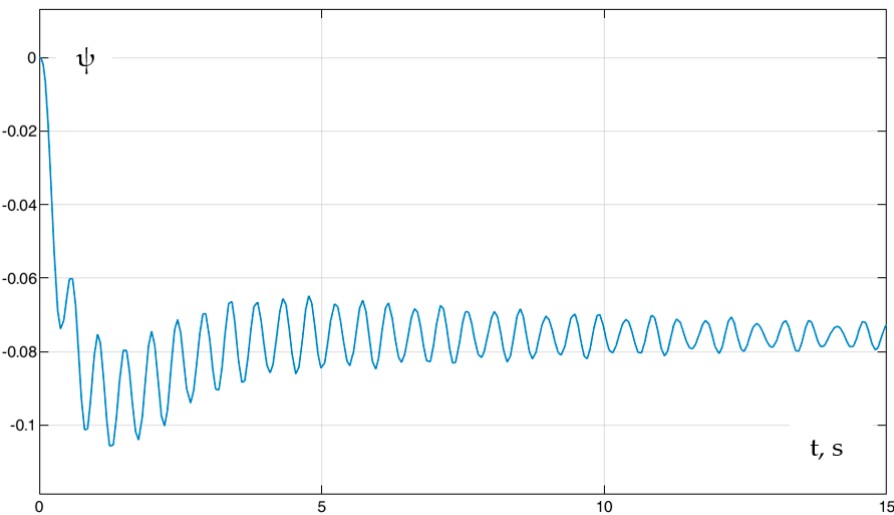

**Figure 15.** The transient process of an elastic link with a UAV mass of 1000 kg and a length of 10 m.

Using the obtained models, we can experimentally solve the problem of finding the first $n$ terms sufficient for an adequate representation of the elasticity model when working with transfer functions $V^\beta(s, x)$ and $V^\delta(s, x)$—transfer functions of the UAV as an elastic link [32].

## 4. Discussion

As noted earlier, despite the existing publications in this area [13–24,33–35], the topic of constructing mathematical models of complex systems remains very actual.

The article discusses stabilization systems for an aeroelastic UAV with digital feedbacks with quantization periods of digital feedback circuits of control signals that are multiples of a certain number—the greatest common divisor of the sampling periods. Following this, we considered the mathematical models of stabilization systems for aeroelastic UAVs.

Of the two existing alternatives—the construction of mathematical models in the time domain and the construction of mathematical models in the field of the Laplace transform—it is proposed to focus on models in the form of Laplace transform. This is because, as indicated in the article, the considered class of systems in the time domain will have a

mathematical model consisting of differential equations and finite difference equations of the type (3). Partial differential equations for the deflection $Z(x,t)$ are also added to them:

$$\frac{\partial^4 Z(x,t)}{\partial x^4} + k^4 \frac{\partial^2 Z(x,t)}{\partial t^2} = R(x,t) \tag{26}$$

where $k^4$–elastic—mechanical constant; $R(x,t)$ is the distribution function of aerodynamic forces.

Finding the solution to this system is a difficult task. The development of methods for analyzing the dynamics and synthesis of the considered class of systems is an extremely difficult task. Thus, an attempt to pass to a description in the field of the Laplace transform is obvious.

In the accepted type of models of the aeroelasticity effect, the use of the Laplace transforms made it possible to obtain a description of the transfer functions of aeroelastic dynamic constraints in a general analytical form—in the class of meromorphic functions. This result is the basis for analyzing the possibilities of approximate representations of aeroelastic dynamic links associated with replacing their transfer functions with the simplest fractional rational ones (expansion in terms of tones of elastic vibrations).

In the general analysis of multidimensional systems, this result allows us to restrict the consideration of complex UAV stabilization systems to a class of systems with rational transfer functions. On this basis, systems with digital feedback loops operating with different quantization periods are further considered. It was assumed that the quantization periods are commensurable (multiplicity to a certain number). This assumption opens up opportunities for equivalent transformations of the model of a multi-rate continuous–discrete system into a model of an impulsive (discrete) single-rate system. The technique of such transformations is rather complicated and can be carried out in an ambiguous form. The article presents the basic model of the considered class of systems in the form of the Laplace transform and indicates the works [23,24] in which the matrix forms of equivalent impulse single-rate systems, convenient for the development of methods for analyzing dynamics and synthesis, are constructed.

In all cases, the impulse representations of the outputs of the equivalent models have the form of rational functions of the indicated variables:

$$y_T^*(s) = f(\psi(e^{sT}),\ \psi_1(e^{sT_1}),\dots,\ \psi_r(e^{sT_r})), \qquad y_{NT}^*(s) = F(\Psi(e^{sT}),\ \Psi_1(e^{sT_1}),\dots,\ \Psi_r(e^{sT_r})).$$

Passing to $z$—transform, we will have the relations:

$$y_T^*(z) = y_T^*(s)\big|_{e^{sT}=z} = f(\psi(z),\ \psi_1(z^{n_1}),\dots,\ \psi_r(z^{n_r})), \tag{27}$$

$$y_{NT}^*(z) = y_{NT}^*(s)\big|_{e^{sT}=z} = F(\Psi(z),\ \Psi_1(z^{n_1}),\dots,\ \Psi_r(z^{n_r})), \tag{28}$$

which define the $z$ transformations of outputs as rational functions $z$. They correspond to the originals:

$$y(kT) = \frac{1}{2\pi j} \oint f(z) z^{k-1} dz, \tag{29}$$

$$y(kNT) = \frac{1}{2\pi j} \oint F(z) z^{kN-1} dz. \tag{30}$$

The problem of the synthesis and analysis of the dynamics of processes in the considered multidimensional multi-cycle continuous–discrete UAV stabilization systems is reduced to the problem of the analysis and synthesis of rational functions $f(z)$ and $F(z)$.

Additionally, the obtained representations of the processes make it possible to enter the frequency characteristics of multi-rate closed systems in a convenient form. Additionally, by introducing certain simplifications, it is possible to build hierarchical mathematical models for solving a particular class of problems.

## 5. Conclusions

This paper presents the results of search studies of complex systems of spatial stabilization of an unmanned aerial vehicle aimed at developing adequate mathematical models for organizing their synthesis and analysis. The complexity of the systems under consideration is characterized by its multiloop, the presence of digital feedback circuits in control loops operating with different sampling periods, and the influence of the elastic properties of an aerodynamic object on the dynamics of control processes. Naturally, the search for ways to form methods of synthesis and the analysis of these systems in the full combination of all elements of complexity appears to be carried out according to the principle "from simple to complex." For this purpose, the corresponding hierarchy of models was defined in the work. It highlights the: 1. Model of a simply connected (one-dimensional) continuous stabilization system with aeroelastic dynamic links; 2. Model of multidimensional continuous systems with ordinary stationary dynamic links (with rational transfer functions); 3. Model of multidimensional continuous–discrete systems with digital parallel feedback circuits, the sampling periods of which are different, but commensurate.

In the class of multi-rate continuous–discrete systems, a model of systems with an arbitrary number of parallel digital feedbacks operating with different sampling periods is proposed. The allocation of a subsystem for quantized variables makes it possible to structure the transformations of a multi-rate system to equivalent models of a single-rate one very efficiently.

By applying the Laplace transform to the analysis of elastic vibrations of the UAV body, the transfer functions of additional dynamic links introduced by elastic deformations into the stabilization loop through gyroscopic devices and accelerometers are obtained. This will make it possible to formulate a methodology for analyzing the influence of aeroelastic constraints on the stabilization loop, which will allow developing approaches to formulate requirements for the effective placement of gyroscopes and accelerometers on the UAV body.

The results obtained in this work provide a basis for creating a fairly complete system of means for theoretical and numerical analysis and the synthesis of complex multidimensional continuous–discrete systems for the stabilization of unmanned aerial vehicles of the considered class.

**Author Contributions:** Conceptualization, V.K.; software, A.K.; validation, A.K.; formal analysis, V.K.; writing—original draft preparation, V.K.; writing—review and editing, V.K. and A.K.; supervision, V.K. and A.K.; project administration, S.D.; funding acquisition, S.D. All authors have read and agreed to the published version of the manuscript.

**Funding:** This work was financially supported by the Ministry of Science and Higher Education of the Russian Federation as part of the project "Fundamentals of Mechanics, Control and Management Systems for Unmanned Aerial Systems with Shape-Building Structures, Deeply Integrated with Power Plants, and Unique Properties Not Used Today in Manned Aviation", No. FEFM-2020-0001.

**Institutional Review Board Statement:** Not applicable.

**Informed Consent Statement:** Not applicable.

**Data Availability Statement:** The data presented in this study are available on request from the corresponding author.

**Conflicts of Interest:** The authors declare no conflict of interest.

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
