# Peer review of "A Mathematical Model for a Conceptual Design and Analyses of UAV Stabilization Systems"

_fluids, doi:10.3390/fluids6050172_

Round 1

Reviewer 1 Report

This paper discusses very relevant topic. In my opinion, the authors pay too much attention to describing methods of increasing stability. As far as it is clear from the article, these methods are known and the authors compare them. It would be better to shorten the description of the methods and just refer to them. At the same time, less attention is paid to the direct comparison of methods. The authors have implemented the methods and conducted experimental research, but the results are only shown, but poorly explained. In addition, the quality of the English language needs to be improved. 

The list of references lacks references to recent research in this area. 

Lines 217-222 are incorrectly formatted, apparently due to the fact that the size of the characters is larger than the main text. The same for lines 246-250. 

Figures 4,5 have a bad quality.

Author Response

Response to Reviewer 1 Comments

Point 1: This paper discusses very relevant topic. In my opinion, the authors pay too much attention to describing methods of increasing stability. As far as it is clear from the article, these methods are known and the authors compare them. It would be better to shorten the description of the methods and just refer to them. At the same time, less attention is paid to the direct comparison of methods. The authors have implemented the methods and conducted experimental research, but the results are only shown, but poorly explained. In addition, the quality of the English language needs to be improved.

 Response 1:  The authors in the article consider the UAV stabilization system as an example of a multidimensional multirate continuous-discrete system. The statement of the problem consists in obtaining a hierarchical set of mathematical models in the area of ​​the Laplace transform or z-transform, which can be used for a wide analysis of the systems and their synthesis, within which it can be, incl. and methods leading to increased stability. Indeed, methods for constructing mathematical models of the systems under consideration have been studied for a long time. But the existing approaches are mainly aimed at analyzing the stability of systems. SISO systems are also considered that do not have cross-links as for MIMO systems. Also, the systems under consideration do not take into account the presence of links that characterize the elastic vibrations of the object. On the construction of equivalent mathematical models, which are suitable for both synthesis and analysis of the frequency characteristics of the system. In their work, the authors only indicated the initial mathematical models, referring to works [23, 24], where the problem of constructing an equivalent mathematical model is solved. Direct comparison of the methods consists in the fact that the authors refuse to use mathematical models in the time domain, such as those that do not make it possible to carry out a frequency analysis of the system, do not allow building structural diagrams of the system and are quite complex, containing differential equations, finite-difference equations, and equations in partial derivatives, describing the vibrations of the body of the apparatus.

Based on your comment, the authors have changed the Discussion section and added a Conclusion section, in which they tried to answer many important questions.

 Point 2: The list of references lacks references to recent research in this area.

Response 2: Even though research in this area began in the 50-the 60s of the 20th century, the issue has not been fully resolved. New articles are coming out. The authors in the article referred to the work on modeling various hybrid systems.

  1. Goorden, M., Dingemans, C., Reniers, M., van de Mortel-Fronczak, J., Fokkink, W., Rooda, J. Supervisory control of multilevel discrete-event systems with a bus structure. Proceedings of 18th European Control Conference, 2019; pp. 3204–3211. doi: 10.23919/ECC.2019.8795835
  2. Guanghao Zhang, Xin Huo, Jinkun Liu, Kemao Ma. Adaptive Control with Quantized Inputs Processed by Lipschitz Logarithmic Quantizer. International Journal of Control Automation and Systems 2011, 19(2), pp. 921-930. doi: 10.1007/s12555-019-0962-z
  3. Chanyoung Ju, Hyoung Il Son. Hybrid systems based modeling and control of heterogeneous agricultural robots for field operations. Proceedings of ASABE Annual International Meeting, 2019, pp. 1–5.
  4. Kramar, V.A.; Volodin, A.N. Mathematical model of the elongated body vibrations to describe the elastic properties of the aerial vehicle. IOP Conf. Series: Materials Science and Engineering 2020, Volume 709 022070, pp. 1-8. doi:10.1088/1757-899X/709/2/022070.
  5. Pann Nu Wai Lin; Nang Lao Kham Longitudinal and Lateral Dynamic System Modeling Of A Fixed-Wing UAV. International Journal of Scientific & Technology Research 2017, Volume 6, Issue 04, pp. 171-174.
  6. López-Briones, Y. F.; Sánchez-Rivera, L. M.; Arias-Montano, A. Aerodynamic Analysis for the Mathematical Model of a Dual-System UAV, Proceedings of the 17th International Conference on Electrical Engineering, Computing Science and Automatic Control, Mexico City; 2020. doi: 10.1109/CCE50788.2020.9299157

Point 3: Lines 217-222 are incorrectly formatted, apparently because the size of the characters is larger than the main text. The same for lines 246-250.

Response 4: Corrected.

Point 3: Figures 4,5 have a bad quality.

Response 4: Corrected. Now these are Figures 2 and 3

Reviewer 2 Report

The article is related to the actual topic of UAV control, but the class of devices under research is not described at the beginning. The presented multiloop continuous-discrete UAV stabilization systems are not new, and figures 1 and 2 do not add scientific value to the article. Some variables and expressions in equations (1) and (3) are not defined, the meaning of (t-0) in discrete control equations is not clear.

The input data and stress signal for the results are not presented, all Simulink models are without any values, obtained results are differently defined in text and figures 12 and 13 ("In Figure 13 red (solid) line"..." the blue (dashed)"), and no evaluation of the results and comparison to others are presented.

 For figures 16 and 17 there is no answer presented why the steady-state differs for the heavier vehicle.

The benefit of the proposed methods should be proven in the discussion by the comparison and numerical evaluation, and conclusions should be formulated.

Author Response

Point 1: The presented multiloop continuous-discrete UAV stabilization systems are not new, and figures 1 and 2 do not add scientific value to the article.

Response 1: Corrected. Figures 1 and 2 were deleted.

Point 2: Some variables and expressions in equations (1) and (3) are not defined, the meaning of (t-0) in discrete control equations is not clear.

Response 2: Corrected. Explanations added to formulas (1) and (3).

Point 3: The input data and stress signal for the results are not presented, all Simulink models are without any values, obtained results are differently defined in text and figures 12 and 13 ("In Figure 13 red (solid) line"..." the blue (dashed)"), and no evaluation of the results and comparison to others are presented.

Response 3: Corrected. Changes made.

Point 4:  For figures 16 and 17 there is no answer presented why the steady-state differs for the heavier vehicle.

Response 4: The steady-state values differ by a very small amount. They can be considered the same. The qualitative indicators of the transition process are changing. Attention is drawn to this.

Point 5: The benefit of the proposed methods should be proven in the discussion by the comparison and numerical evaluation, and conclusions should be formulated.

Response 5: Corrected. Based on your comment, the authors have changed the Discussion section and added a Conclusion section, in which they tried to answer many important questions.

Round 2

Reviewer 2 Report

The sentence in Russian in the discussion section should be removed or translated.

The used term "tact T" is not clear, the discretization or sampling period can be used.

Pictures should be improved to make the text readable and not oversized.

Author Response

Point 1: The sentence in Russian in the discussion section should be removed or translated.

Response 1: Corrected. The sentence in Russian has been translated.

Point 2: The used term "tact T" is not clear, the discretization or sampling period can be used.

Response 2: Corrected. The term “Sampling period” was used.

Point 3: Pictures should be improved to make the text readable and not oversized.

Response 3: Corrected. Pictures have been corrected
